# Progress in the Development of Biosensors Based on Peptide–Copper Coordination Interaction

**DOI:** 10.3390/bios12100809

**Published:** 2022-09-30

**Authors:** Gang Liu, Ning Xia, Linxu Tian, Zhifang Sun, Lin Liu

**Affiliations:** 1College of Chemistry and Chemical Engineering, Anyang Normal University, Anyang 455000, China; 2College of Chemistry and Chemical Engineering, Henan University of Technology, Zhengzhou 450052, China

**Keywords:** peptide, copper ion, fluorescent assay, colorimetry, electrochemical biosensor

## Abstract

Copper ions, as the active centers of natural enzymes, play an important role in many physiological processes. Copper ion-based catalysts which mimic the activity of enzymes have been widely used in the field of industrial catalysis and sensing devices. As an important class of small biological molecules, peptides have the advantages of easy synthesis, excellent biocompatibility, low toxicity, and good water solubility. The peptide–copper complexes exhibit the characteristics of low molecular weight, high tenability, and unique catalytic and photophysical properties. Biosensors with peptide–copper complexes as the signal probes have promising application prospects in environmental monitoring and biomedical analysis and diagnosis. In this review, we discussed the design and application of fluorescent, colorimetric and electrochemical biosensors based on the peptide–copper coordination interaction.

## 1. Introduction

Copper is an essential trace metal element for redox chemistry and the growth and development of organic organisms. In the human body, copper is the third largest trace element besides iron and zinc, which plays an important role in human metabolism. Copper ions are involved in the synthesis of many enzymes, such as cytochrome oxidase, peroxide dismutase, tyrosinase, ascorbic acid oxidase, lysyl oxidase, and ceruloplasmin [1]. As the active centers of enzymes, copper ions participate in many electron transfer and redox reactions involving oxygen in various biological systems and life processes. However, they exhibit great toxicity when their content exceeds the inherent steady-state regulation range of organisms [2]. In organisms, copper ions mainly exist in the form of complexes with proteins or peptides, although small molecule-based metal chelators can be used to maintain the concentration of free copper ions below picomolar levels. In peptide–copper complexes, the terminal amino group, deprotonated amide nitrogen and various side chain donors are the binding sites of copper ions [3,4,5,6,7,8,9]. Histidine residues in the peptide sequence, especially in the first three positions of the N-terminal (denoted as His1, His2, and His3), are the main sites for the binding of copper ions. For example, the His1-contained peptide can bind Cu^2+^ by two nitrogen atoms in the free amine of the N-terminal and the imidazole group of histidine residue [3,9], the His2-contained peptide can bind Cu^2+^ by three nitrogen atoms in the free amine of N-terminal, the second amide as well as the imidazole group of histidine residue [6], and the His3-contained peptide, also denoted as the amino-terminal copper and nickel (ATCUN) binding peptide, can chelate Cu^2+^ and Ni^2+^ ions with femtomolar affinity. Cu^2+^ is coordinated by ATCUN peptide through the N-terminal free amine, the second and third amides, and the imidazole group of histidine residue [6,7,8,9]. In general, the location of histidine in the peptide sequence has a decisive impact on the characteristics of peptide–copper complexes, including the coordination mode, binding affinity, redox potential, and catalytic activity. In addition, divalent copper ions are paramagnetic and easy to combine with electron chelating groups with appropriate cavity volume. This will cause the change of electron or energy transfer mode when the organic molecules are excited, resulting in the reduction or quenching of the fluorescence signal [10,11,12,13,14]. The design of biosensors based on the intrinsic fluorescence quenching ability and the redox activity of copper ions has been widely covered by researchers.

Previously, several review papers have overviewed the design of peptide-derived biosensors by using peptides as the recognition elements or enzyme substrates [15,16,17,18,19]. However, there is little overview of the development of biosensors based on the formation of peptide–copper complexes. In this work, we, for the first time, review the progress in the design of fluorescent, colorimetric, and electrochemical biosensors based on the peptide–copper coordination interaction (Figure 1). In view of the fluorescence quenching of Cu^2+^ by electron or energy transfer, dye-labeled peptide monomers or their assemblies have been designed to develop fluorescent biosensors for the sensitive detection of metal ions and proteases. Due to the specific coordination mode, copper ions can regulate the aggregation of plasmonic metal nanoparticles alongside with the change of color solution. Meanwhile, the peptide–copper complexes with catalytic properties can mediate the in situ growth of metal nanoparticles and accelerate the chromogenic reaction, which facilitate the design of colorimetric biosensors. In electrochemical biosensors, the peptides can be modified on the electrode surface to determine copper ions. More interestingly, the peptide–copper complexes with electrocatalytic ability could be used as the signal tags for the design of novel electrochemical biosensors.

## 2. Fluorescent Biosensors

### 2.1. Peptide Fluorescent Probes for Copper Ions

Traditional methods for the detection of metal ions mainly include atomic spectrometry, inductively coupled plasma, chromatography, and mass spectrometry. Although these methods can achieve accurate determinations, they require the use of large and expensive instruments with complex operation procedures. As an important class of homogeneous detection methods, fluorescent probes have widely used in the field of bioassays in vitro and in vivo. Peptides have been widely used to develop various biosensors due to their characteristics of easy synthesis, good biocompatibility, and low toxicity [15,17,18]. Moreover, their rich structural composition and variable structural characteristics are favorable to designing sequence-specific peptides for the selective coordination with various analytes, including metal ions. Based on the type of fluorescent groups, peptide-based fluorescent biosensors can be divided into endogenous fluorophores-based and exogenous fluorophores-labeled biosensors. Several amino acids, such as tryptophan (Trp), tyrosine (Tyr), and phenylalanine (Phe), exhibit endogenous fluorescence [20,21,22,23,24]. The peptide of HDSGWEVHH has been used as a label-free fluorescent probe for the detection of Cu^2+^ and S^2−^ [23]. However, the fluorescence intensity of endogenous fluorophores is low and the sensitivity and selectivity of such methods are poor for the detection of copper ions. Therefore, exogenous fluorescent molecules or metal nanoclusters (NCs) are often used to modify peptides, including dansyl (Dns), carboxyl fluorescein (FAM), fluorescein isothiocyanate (FITC), and pyrene sulfonyl chloride (PySO_2_) (Table 1). Among them, dansyl is a commonly used fluorescent group to modify peptide for the detection of metal ions [25,26,27,28,29,30,31,32,33,34]. Meanwhile, the sulfonamide group in it can facilitate the formation of peptide–metal complexes [35]. Regarding this, Wang et al. reported a peptide fluorescent probe (dansyl-Ser-Pro-Gly-His-NH_2_) for the detection of Cu^2+^ in 100% aqueous solution [29]. Histidine (His) and serine (Ser) separated by the spacer (Pro-Gly) are the binding sites to coordinate with Cu^2+^. After being complexed by the peptide, Cu^2+^, with a paramagnetic property, quenched the fluorescence of the dansyl group near the binding center via electron or energy transfer from the excited fluorophore to the coordinated Cu^2+^. In addition, metal NCs as the fluorescent reporters have attracted extensive attention because of their promising properties (e.g., easy synthesis, good biocompatibility, and high stability) [36,37,38]. In order to evaluate the selectivity and anti-interference of the sensor, other common metal ions (K^+^, Na^+^, Mg^2+^, Fe^2+^, Cu^+^, Co^2+^) and plasma were determined as the control experiments. The results indicated that these species showed negligible interference for Cu^2+^ detection. Finally, tap water was analyzed as a real sample with excellent results. Qian et al. developed a Cu^2+^ biosensor by using the peptide of lipoic acid-Arg-Gly-Asp-Asp-Lys-lipoic acid (LA-RGDK-LA) as the templet and ligand to synthesize AuNCs [37]. Zhuang et al. reported the detection of Cu^2+^ with the peptide (CCYWDAHRDY)-modified AuNCs, in which bovine serum albumin (BSA) was used as the template for the synthesis of fluorescent AuNCs [38]. The binding of Cu^2+^ ions to the peptide-modified AuNCs caused the fluorescence quenching.

Although many peptide-based sensors for Cu^2+^ detection have been developed, most of them work in the “signal-off” formation and suffer from the intrinsic fluorescence background of biomolecules in biological media. To solve this problem, Falcone et al. designed two Tb^3+^-based luminescent ATCUN peptides for reversible and “turn-on” detection of Cu^2+^ in biological media [39,40]. The ATCUN motif in the probe exhibited high affinity toward Cu^2+^. The Tb^3+^-DOTA (1,4,7,10-tetraazacyclododecane-1,4,7,10-tetraacetic acid) complex was sensitized to the Trp residue via the “antenna effect”, providing a long-lifetime luminescence. After the formation of ATCUN–Cu^2+^ complex, the luminescence of both Trp and Tb^3+^ was quenched. In addition, Jung et al. developed a dansyl-labeled tripeptide fluorescent probe (DNS-LLC) to detect Cu^+^ for the first time [41]. This probe formed a 1:1 complex with Cu^+^ in aqueous solution and showed a “signal-on” response to Cu^+^. The position of cysteine residue in the peptide chain showed a significant effect on the selectivity. The DNS-LLC could penetrate live RKO cells and monitor Cu^+^ in Golgi matrix with a “signal-on” fluorescent response.

Multiple metal ions can also be determined by one peptide probe by reasonable design of the peptide sequence. For example, Pang et al. reported that dansyl-His-Pro-Gly-Trp-NH_2_ with a dansyl fluorophore and a Trp residue exhibited a different response toward Cu^2+^ and Hg^2+^ [27]. After the coordination of peptide with Hg^2+^, the distance between the Trp residue (donor) and the dansyl fluorophore (acceptor) was shortened, increasing the efficiency of the fluorescent resonance energy transfer (FRET). As a result, the fluorescence intensity of Trp at 355 nm decreased and that of dansyl at 550 nm increased with a 45 nm blue shift. The fluorescent probe showed good selectivity and sensitivity toward Hg^2+^ and Cu^2+^ among 17 metal ions and 6 anions in N-2-hydroxyethylpiperazine N-2-ethane acid (HEPES) buffer solution. Moreover, Wang et al. synthesized a peptide fluorescent probe (dansyl-Gly-Cys-NH_2_, DGC) for the selective detection of Cu^2+^ and Cd^2+^ in aqueous solution and within live LNCaP cells [32]. At the same time, they designed and synthesized two dansyl-labeled fluorescent peptide probes (dansyl-Ser-Pro-Gly-His-NH_2_ and dansyl-His-Pro-Gly-Glu-NH_2_) for the “signal-on” detection of Zn^2+^ and “signal-off” detection of Cu^2+^ through two different mechanisms [30,33]. The detection limits for Zn^2+^ and Cu^2+^ are as low as 4.9 nM and 15 nM, respectively.

### 2.2. Peptide-Cu^2+^ Complexes for Fluorescent Detection of Anions

The formed peptide–Cu^2+^ complexes can be further used to detect other substances that can bind to Cu^2+^ with high binding constant. Thus, peptides can be employed as the dual-functional probes to develop “on–off–on” biosensors for multiple analytes [28,29,34,42,43,44,45,46,47,48,49]. For example, anions, such as S^2−^ and CN^−^, can sequester Cu^2+^ from the peptide–Cu^2+^ complexes, leading to the restoration of the quenched fluorescence. Recently, Hu et al. designed a dansyl-labeled hexapeptide for the detection of Cu^2+^ and S^2−^ [34]. The peptide formed a 1:1 complex selectively with Cu^2+^ without interference by other metal ions, resulting in the fluorescence quenching. After the addition of S^2−^, Cu^2+^ was released from the complex and the fluorescence was recovered. Finally, the dansyl-labeled peptide was successfully utilized for the fluorescence imaging of Cu^2+^ and S^2−^ in living HeLa cells and zebrafish. Wang’s group used a series of fluorophore-labeled peptides to detect Cu^2+^ and S^2−^ or H_2_S by both colorimetric and fluorescent methods [29,44,45,46]. For instance, FITC-Ahx-Ser-His-NH_2_ was synthesized to continuously monitor Cu^2+^ and S^2−^ in buffer solution and within living cells [45]. The probe showed high sensitivity for colorimetric analysis of Cu^2+^, and the color change from yellow to orange was observed with naked eye, reflecting the fluorescence “signal-off” response to Cu^2+^. Then, the peptide-Cu^2+^ complex was used for colorimetric and fluorescent analysis of S^2−^. The system could continuously and alternately determine Cu^2+^ and S^2−^ with good reversibility. Moreover, Hao et al. designed and synthesized a fluorescent probe of FITC-Ahx-His-Glu-Phe-Cys-NH_2_ [47]. Histidine and cysteine residues in the peptide chain acted as the Cu^2+^-binding groups. Through photon-induced electron transfer (PET), the coordination between Cu^2+^ and peptide caused the electron transfer, which led to the fluorescence quenching and realized the specific detection of Cu^2+^. The addition of S^2−^ caused the removal of Cu^2+^ from the system and led to the fluorescence recovery. By mimic the binding mode of the ceruloplasmin for Cu^2+^, Wang et al. reported a peptide-based multifunctional probe (FITC-Ahx-Gly-His-Lys-NH_2_) for the fluorescence detection of Cu^2+^ and H_2_S in live colon cancer cells [46]. Other metals did not interfere with the ability of the probe to detect Cu^2+^ ions. In addition, five common soluble copper salts were tested for anionic interference. The results showed that the anions had no obvious interference with Cu^2+^ ions. Meanwhile, they synthesized a FITC-labeled peptide ligand (FITC-Ahx-Ser-Pro-Gly-His-NH_2_), in which His and Ser residues were the coordination sites for Cu^2+^ [44]. The peptide–Cu^2+^ complex was further used to detect H_2_S in living cells and zebrafish larvae. Taking advantage of the capability of histidine residue to coordinate with Zn^2+^ and Cu^2+^, Wang et al. synthesized a peptide probe (dansyl-His-Lys-His-dansyl) with lysine and histidine residues as the main chain for the detection of multiple analytes in live cells [28]. The binding of histidine residue and Zn^2+^ inhibited the photo-induced electron transfer (PET) from the indole moiety to the dansyl fluorophore, generating a “turn-on” fluorescence response. Conversely, the complexation of Cu^2+^ quenched the fluorescence and resulted in a “turn-off” response. The detection of S^2−^ was further achieved through the formation and separation of CuS. The probe showed good cell permeability, low biotoxicity, and high selectivity as well as sensitivity.

Cyanide can impair the cellular functions of metalloenzymes by binding with the metal cofactors. Thus, cyanide is considered to be one of the most toxic anions, even at a very low concentration. Jung et al. designed a peptide-based system for the detection of cyanide in aqueous solution based on the ATCUN motif, in which the tripeptide Ser-Ser-His (NBD-SSH) was labeled by 7-nitro-2,1,3-benzoxadiazole at the N-terminal (Figure 1) [48]. The formation of the NBD-SSH–Cu^2+^ complex resulted in the color change of the probe from yellow to orange, which was accompanied by the fluorescence quenching. The addition of CN^−^ successfully removed Cu^2+^ from the complex. As a result, the solution color turned yellow, and the fluorescence signal was enhanced. This method realized the highly sensitive detection of CN^−^ through the colorimetric and fluorescent analysis. This method is the first example of using a peptide-integrated system to sensitively detect cyanide in 100% aqueous solution. Other anions, such as F^−^, Cl^−^, Br^−^, I^−^, NO^3−^, SCN^−^, AcO^−^, HCO^3−^, SO_4_^2−^, ClO^4−^, HPO_4_^2−^, and HAsO_4_^2−^, showed no significant effect on the colorimetric detection of cyanide. Recently, Wang et al. reported a bifunctional peptide probe (FLH) based on fluorescence “on-off” response and colorimetric analysis [43]. In this method, FLH could detect Cu^2+^ quickly and selectively with a detection limit as low as 1.5 nM, and the formed FLH-Cu complex showed a colorimetric change and a “signal-on” response to CN^−^ with a detection limit of 12.7 nM. Interference experiments showed that other ions did not induce the change of solution color from light yellow to orange under natural light, and the effect of other ions on the fluorescence intensity was negligible. Finally, the probe was successfully applied to monitor Cu^2+^ and CN^−^ in living cells through fluorescence imaging.

### 2.3. Other Fluorescent Biosensors Based on Peptide-Cu^2+^ Complexes

Based on the high binding affinity of ATCUN peptide and Cu^2+^ and the high inherent quenching ability of Cu^2+^ toward the fluorophore, Deng et al. proposed a single-labeled peptide probe for the detection of protease (Figure 2) [50]. In this work, two proteases, caspase-3 and β-secretase, were determined in a “signal-on” and “signal-off” format, respectively. For the detection of caspase-3, the Dns group was labeled at the C-terminal of peptide. Cu^2+^ could be coordinated by the peptide probe SGHDEVDK-Dns through the interaction with the ATCUN motif (SGH) at the N-terminal, thus leading to the fluorescence quenching. Once the peptide was catalytically cleaved by caspase-3, the released fragment of K-Dns exhibited high fluorescence signal even in the presence of Cu^2+^. For the assay of β-secretase, the fluorophore was labeled at the N-terminal of peptide. The probe EVNLDAHFWADK-Dns had a low binding affinity toward Cu^2+^ and showed a high fluorescence signal. Once the probe was cleaved by β-secretase, Cu^2+^ would be coordinated by the ATCUN moiety (DAH) in the released DAHFWADK-Dns segment, thus leading to the fluorescence quenching.

Conventional small-molecule organic dyes may confront the problem of aggregation-caused quenching and photobleaching. To overcoming this disadvantage, organic luminaries with aggregation-induced luminescence (AIE) characteristics have attracted extensive attention in recent years [51,52]. In the dispersed state of AIE molecules, the vibration and rotation inside the molecules lead to the rapid non-radiative attenuation of the excited state. Thus, the fluorescence of AIE molecules in dilute solution is very weak. However, in the aggregated state, the non-radiative attenuation of restriction of intra-molecular rotation (RIR) is suppressed, resulting in the strong fluorescence. Liu et al. reported an AIE-based probe (TPE-Py-EEGTIGYG) for the highly selective detection of Cu^2+^ in aqueous solution and living cells [53]. The peptide of EEGTIGYG exhibited good water solubility and specific binding ability toward cell membrane. The TPE-Py-EEGTIGYG monomers could self-assemble into aggregates at a high concentration (25 μM) in aqueous solution and thus produce a strong fluorescence signal. Cu^2+^ could be selectively and sensitively determined by quenching the fluorescence of nanoaggregates. Interestingly, TPE-Py-EEGTIGYG at a low concentration (5 μM) exhibited weak fluorescence intensity in aqueous solution. When it was anchored to the cell membrane, the fluorescence was significantly enhanced. Thus, the probe can be used to determine Cu^2+^ entering and leaving the cell. Moreover, Gour et al. developed a pyridothiazole-based AIE probe for the in vitro detection of amyloid peptide [54]. The probe of 4-(5-methoxythiazolo [4,5-b] pyridin-2-yl) benzoic acid (PTC1) shows AIE characteristics. However, in the presence of Cu^2+^, PTC1 molecules were dissociated or destroyed, resulting in the fluorescence quenching. The coordination of Cu^2+^ by amyloid peptide caused the significant recovery of fluorescence signal. Cai et al. reported the detection of Cu^2+^ and caspase-3 based on the self-assembly of 4-nitro-2,1,3-benzoxadiazole (NBD)-labeled peptides with the sequence of NBD-FFYEEGGH and NBD-FFFDEVDGGH (Figure 3) [55]. The peptides could self-assemble into nanofibers with enhanced cellular uptake and brighter fluorescence. The coordination of Cu^2+^ to the nanofibers by the interaction between Cu^2+^ and GGH segment caused the fluorescence quenching, which allowed for the determination of Cu^2+^ in cells. Furthermore, the NBD-FFFDEVDGGH–Cu^2+^ nanofibers were used to determine caspase-3 activity in vitro and within cells. Upon the enzymatic cleavage of the peptide between NBD-FFFDEVD and GGH, the Cu^2+^–GGH complex was removed, resulting in the fluorescence recovery. Moreover, Kim et al. prepared two types of pyrene-labeled peptide nanofibrils for “light-off” and “light-up” fluorescent detection of Cu^2+^ and Ag^+^ in HeLa cells, respectively [56]. The peptides consist of a hydrophobic linear- or branched-alkyl chain and a hydrophilic histidine-rich peptide (HGGGHGHGGGHG). The nanofibrils were further used as the template scaffolds for the preparation of monodispersed and highly stable silver nanoparticles (AgNPs).

## 3. Colorimetric Biosensors

### 3.1. Aggregation and Growth of Metal Nanoparticles Mediated by Peptide–Cu^2+^ Complexes

Colorimetric biosensors with apparent color changes are particularly attractive because of their high sensitivity, minimum instrumental investment and visual signal-readout [57]. Gold nanoparticles (AuNPs) are the commonly used indicators for colorimetric assays due to their good stability and high extinction coefficient. The dispersed AuNPs with small diameters usually exhibit wine red in the solution. However, when AuNPs are subjected to aggregation, the color changes from red to blue due to the transfer of surface plasmon absorption to a longer wavelength. Peptides and proteins can be rapidly adsorbed or assembled on the surface of AuNPs to trigger or prevent the aggregation of AuNPs. Based on the coordination of Cu^2+^ with peptide, AuNPs-based colorimetric methods have been used for the quantification of peptide and Cu^2+^ (Table 2) [58,59,60,61]. For example, Wang et al. investigated the interaction between amyloid-β peptides and Cu^2+^/Zn^2+^ ions using AuNPs as the indicators [59]. Zhou et al. reported the AuNPs-based colorimetric assay of Aβ(1–40) (one of the biomarkers of Alzheimer’s disease) by the interaction between Aβ(1–40) and Cu^2+^, in which the coordination of Cu^2+^ with Aβ(1–40) induced the aggregation and color change of the Aβ(1–40)-AuNPs conjugates (Figure 4) [60]. The detection limit of this method for Aβ(1–40) was found to be 0.6 nM. Moreover, Pelin et al. found that Cu^2+^ could induce the aggregation of amphiphile peptide-modified AuNPs, thus achieving the colorimetric detection of Cu^2+^ in aqueous solution with a detection limit of 0.19 μM [61].

In contrast to the AuNPs-based assays, AgNPs-based colorimetric analysis exhibits the advantages of lower cost and higher extinction coefficient. For this view, Ghodake et al. reported the colorimetric detection of Cu^2+^ by using casein peptide-functionalized AgNPs as the indicators [62]. Hu et al. developed a colorimetric immunosensor for the specific detection of Aβ_(1–40/1–42)_ by using antibody-modified AgNPs (Ab-AgNPs) as the indicators and Cu^2+^ ions as the linkers [63]. The coordination between Cu^2+^ and Aβ_(1–40/1–42)_ captured by the Ab-AgNPs induced the aggregation of AgNPs, which was accompanied by the color change from yellow to red. The method allowed for the determination of Aβ_(1–40/1–42)_ with a detection limit down to 86 pM.

Peptides with cysteine and positively charged amino acid residues can induce the aggregation of AuNPs through the Au–S and electrostatic interactions. The cleavage of the peptide by protease may prevent the aggregation of AuNPs, thus facilitating the colorimetric detection of protease [64]. However, in the aggregation-based colorimetric assays, the peptide substrate must be well-designed for interacting with gold surface. Recently, Liu and coworkers reported a colorimetric method for protease detection based on the in-situ growth of AuNPs and the peptide-mediated catalytic activity of Cu^2+^ (Figure 5) [65]. In this method, AuNPs were generated by the reduction of chloroauric acid with AA as the reducing reagent, but the Cu^2+^-catalytic oxidation of AA limited the formation of AuNPs. Because the peptide substrate and the proteolytic products exhibited different binding affinity toward Cu^2+^, the oxidation of AA and the formation of AuNPs were mediated by proteolytic reaction. Thus, the color change was dependent upon the concentration and activity of protease. To exclude the interference from matrix components in biological fluids, the peptide substrates were modified on the surface of magnetic microbeads. The cleavage of the peptides on the surface of magnetic microbeads by protease led to the generation or detachment of Cu^2+^-binding ATCUN peptide segments. This sensing strategy is fundamentally different from the aggregation-based colorimetric assays, thus providing a new concept for the design of protease biosensors.

### 3.2. Peptide-Cu^2+^ Self-Assembles as the Signal Tags

Phenylalanine residues in the peptide chains can promote the self-assembly of peptides into various molecular conformations via hydrogen-bonding and π-stacking interactions [66,67]. The histidine residues play an important role in the coordination of divalent metal ions as the active centers of many natural enzymes [1,68]. Thus, incorporation of metal ions with peptides may grant for assembles with catalytic activity or new functionality. Sun et al. prepared the biotinylated peptide-Cu^2+^ nanoparticles (Cu-P NPs) by “one-pot” self-assembly with biotin-FFKGH as the precursor (Figure 6) [69]. The Cu–P NPs contained extensive active center of Cu^2+^ ions. Under acidic conditions, Cu^2+^ ions with intrinsic peroxidase activity could be released to promote the catalytic oxidization of 3,3′,5,5′-tetramethylbenzidine (TMB). The Cu–PNPs were further used as the labels for the colorimetric immunoassays of prostate specific antigen (PSA). The target PSA attached on the ELISA was recognized by the streptavidin-conjugated secondary antibody (Ab_2_–SA), allowing for the capture of Cu–PNPs by the streptavidin–biotin interaction. Then, the Cu^2+^ ions were released to catalyze the oxidation of the colorless TMB into blue oxTMB in the presence of H_2_O_2_. The biosensor has a linear range of 0.001–1 ng/mL with a detection limit of 1 pg/mL. In addition, Qi and coworkers synthesized a laccase-like nanozyme by using Cu^2+^ and dipeptide Cys-His (CH) as the precursors [70]. The CH–Cu nanozyme formed thorough the coordination interaction between Cu^2+^/Cu^+^ and CH exhibited high catalytic activity and thermostability even at extreme pH and high temperature. It has been successfully used for the degradation of chlorophenol and bisphenol and the determination of epinephrine by a smart phone. Zhang et al. prepared various self-assembles with histidine-containing peptides as the precursors [71]. The peptide conformation changed from random coil to β-sheet structure and nanotube with hydrolase activity. After integration with Cu^2+^, the morphology of self-assembles was altered and the resulting hybrid nanomaterials showed excellent peroxidase-like activity. These works suggest that artificial enzymes could be designed with histidine-containing self-assemble peptides by incorporation of copper ions.

## 4. Electrochemical Biosensors

### 4.1. Peptide-Modified Electrodes for Copper Detection

A variety of optical, electrochemical, and mass biosensors have been developed for the detection of metal ions with peptides as the probes. Among them, electrochemical biosensors have the characteristics of easy miniaturization, rapid response, and high sensitivity. Therefore, it is of great significance to design electrochemical biosensors for real-time detection of copper ions. Compared with other biological macromolecules, peptides have the characteristics of small molecular weight, are easily modified, and have a simple synthesis procedure. The peptide-modified electrodes have been prepared to detect copper ions, including the simplest ATCUN tripeptide Gly-Gly-His (GGH) [72,73,74,75,76,77]. For example, Yang et al. prepared an electrochemical biosensor for rapid detection of Cu^2+^ by modification of tripeptide GGH onto 3-mercaptopropionic acid (MPA)-modified gold electrode via carbodiimide-mediated amine coupling reaction [72]. The redox behavior of GGH–Cu^2+^ complexes on the electrode surface has been investigated by cyclic voltammetry. In order to test the performance of the modified electrode for the analysis of real samples, copper ions in diluted tap water and lake samples were determined. The concentration of Cu^2+^ in tap water was 0.27 ppm, which was consistent with that (0.33 ppm) determined by inductively coupled plasma mass spectrometry (ICP-MS). Subsequently, they reported the direct synthesis of tripeptide GGH on the gold electrode surface for the binding of Cu^2+^ [73]. The covalent coupling of Gly, Gly, and His through three steps on the MPA self-assembled monolayers (SAMs)-covered electrode surface was confirmed by characterizing the electrochemically desorbed peptide with mass spectrometry. Compared with previous reports, the direct synthesis of GGH on the electrode surface effectively provided the recognition site of Cu^2+^ and thus improved the detection sensitivity. However, the SAMs formed through the Au–S covalent bonds on the gold surface may be affected by temperature, thus reducing the stability of biosensors. In order to solve this problem, Gooding’s group used aryldiazonium salt to modify the glassy carbon electrode and micro planar electrode for the immobilization of Cu^2+^-binding peptide [74,75]. In addition, the strategy of covalent immobilization of peptides through silicon–carbon bonds has attracted much attention. Sam et al. prepared a Cu^2+^ biosensor by covalent modification of tripeptide GGH on the surface of porous silicon electrode through the silicon-carbon interaction [76]. These methods solved the problem of instability of SAMs on the gold surface, making the commercialization of Cu^2+^ biosensors possible.

The binding of Cu^2+^ with peptide on the gold surface may induce the change in the peptide conformation and electron conductance of electrode interface [78]. For example, Chen et al. used surface plasmon resonance (SPR) spectroscopy to study the conformational change of Cu^2+^-binding peptide of Ac-CGGGSIRKLEYEIEELRLRIG-NH_2_ [79]. After the injection of Cu^2+^ onto the peptide-modified gold chip, the SPR angle increased significantly, indicating that Cu^2+^ was successfully captured by the peptide. Electrochemical impedance spectroscopy (EIS) can measure the electrical resistance of the metal/solution interface and is capable of delivering measurable signal change resulting from the difference in analyte concentration. This technique has been used to achieve the highly sensitive detection of Cu^2+^ ions with peptide-modified electrodes. For example, Jiang et al. developed an EIS biosensor for Cu^2+^ detection with GGH-covered AgNPs-modified glassy carbon electrode [80]. In the absence of Cu^2+^, the charge transfer resistance (R_ct_) between the peptide-modified electrode and the [Fe(CN)_6_]^3−/4−^ probe is high. However, the value decreased significantly after Cu^2+^ ions were captured by the sensing electrode. In addition, Mervinetsky et al. prepared a Cu^2+^ biosensor based on the impedance change by covalently coupling of lipoic acid-functionalized tripeptide GGH to the gold electrode [81]. With the increase of Cu^2+^ concentration, the conformation of peptide on the electrode surface changed and the SAMs became more and more dense, thus resulting in the significant change of R_ct_ value.

### 4.2. Electrochemical Biosensors with Peptide-Cu^2+^ Complexes as the Signal Reporters

ATCUN peptide is characterized by a free amine group at the N-terminus and a histidine residue in the third position. Cu^2+^ was coordinated by the free amine, the second and third amides, and the imidazole group of histidine residue [6,7,8,9]. In actual fact, when the ATCUN peptide was anchored by the covalent coupling reaction between the free N-terminus amine group and the activated carboxyl group on the electrode surface, the binding mode and redox activity of the immobilized peptide–Cu^2+^ complex might be essentially different from the ATCUN–Cu^2+^ complex [82,83]. Interestingly, Deng et al. investigated the redox activity of various ATCUN–Cu^2+^ complexes and found that they exhibited an electrocatalytic ability for water oxidation even at neural pH and low oxidation potential (Figure 7) [84]. However, the acetylated ATCUN peptide did not significantly change the redox property of Cu^2+^. Based on this fact, nanocatalysts were prepared by modifying the ATCUN–Cu^2+^ complexes on the surface of AuNPs by the Au–S bonds and then used as the signal labels to develop a sandwich-like biosensor for the detection of DNA. The analytical performances were shown in Table 3. This is the first electrochemical biosensor by using H_2_O molecule as the catalytic substrate. Furthermore, Deng et al. developed a “signal-on” electrochemical biosensor for the detection of caspase-3 activity by the formation of ATCUN peptide on the electrode surface [85]. In this work, the peptide substrates specific to caspase-3 were anchored on the graphene electrode surface through the hydrophobic and π-stacking interactions. The enzymatic hydrolysis of peptide substrates led to the generation of ATCUN motif. The formed ATCUN–Cu^2+^ complexes on the electrode surface exhibited high electrocatalytic ability for water oxidation. As a result, caspase-3 has been determined at the concentration range of 0.5 pg/mL~2 ng/mL with a detection limit of 0.2 pg/mL. The biosensor was further used to evaluate drug-induced cell apoptosis with satisfactory results. Meanwhile, Liu’s group developed an electrochemical immunosensor by using protease as the signal reporter to catalyze the generation of Cu^2+^-binding ATCUN peptide [86]. In this work, carbon nanotubes (CNTs) were used as the carriers to load the recognition antibody and protease for signal amplification. Protease on the surface of CNTs promoted the generation of ATCUN peptides by catalyzing the cleavage of the substrates. The resulting ATCUN–Cu^2+^ complexes showed good electrocatalytic property toward water oxidation, thus leading to the second signal amplification through the redox cycling reaction.

Recently, we studied the redox activity of the Cu^2+^ complexes formed with the His1- and His2-containing peptides [87]. It was found that the Cu^2+^ complexes formed with His1-containing peptides exhibited excellent catalytic activities for the electrochemical reduction of oxygen and chemical oxidation of ascorbic acid (AA) by O_2_. However, the His2-containing peptides were able to coordinate with Cu^2+^ ions and depress the catalytic activity of Cu^2+^. Finally, the Cu^2+^ complexes formed with His1-containing peptides were modified on the surface of AuNPs to produce nanocatalysts for oxygen reduction or AA oxidation. The resulting nanocatalysts were further used as the signal labels for the design of electrochemical and fluorescent biosensors with PSA as the model target (Figure 8). In the electrochemical immunoassays, the nanocatalysts promoted the electrocatalytic reduction of oxygen. In the fluorescent immunoassays, the nanocatalysts catalyzed the oxidation of AA, and the oxidization product of dehydroascorbic acid (DHA) was then reacted with o-phenylenediamine (OPD) to produce fluorescent 3-(dihydroxyethyl)furo [3,4-b]quinoxaline-1-one (DFQ). The electrochemical and fluorescent methods show a detection limit of 0.40 and 1.00 pg/mL, respectively. Meanwhile, it was found that the Cu^2+^ complexes performed with the His1-containing peptides reveal high activity for the catalytic oxidation of OPD to produce fluorescent 2,3-diaminophenazine (OPDox) and maintained excellent selectivity for PSA detection even in the presence of high concentration of interfering proteins [88]. Based on the difference in the redox potential and current of Cu^2+^ before and after being coordinated by the His1- or His3-containing peptide, Feng et al. proposed a general, label-free, and homogeneous electrochemical biosensor for the detection of protease activity [89]. The angiotensin-converting enzyme (ACE) and thrombin were tested as the analyte examples. In this work, ACE catalyzed the hydrolysis of peptide substrates to produce His1-containing peptides and thrombin promoted the generation of His3-containing peptides. The resulting peptide–Cu^2+^ complexes exhibited a couple of redox peaks, which are essentially different from the free Cu^2+^ ions or the substrate–Cu^2+^ complexes.

## 5. Conclusions

In summary, this work provided an overview on the advancement of different biosensors based on the peptide–copper complexes, including fluorescence, colorimetric, and electrochemical methods. Because of their advantages of easy preparation, high binding affinity, and good biocompatibility, the peptide–copper complexes have been broadly employed in diverse fields of environment monitoring, food safety, and disease diagnosis. For example, copper ions can be sensitively detected by fluorescent peptide probes through quenching the endogenous fluorescence of amino acid residues or the exogenous fluorescence of dyes labels due to its paramagnetic capability. The peptide–copper complexes with excellent catalytic and redox properties have been elaborately used as signal labels to construct colorimetric and electrochemical sensing platforms.

Although some important progress has been made, there are still some problems to be further explored. For example, novel copper complexes with more enzyme-mimicking abilities (e.g., laccases and glucose oxidase) are desired and the efficiency and specificity toward substrates should be further improved. The binding and catalytic mechanism of peptide–copper complexes need to be further studied under uniform standards and experimental conditions, because experimental parameters have a significant influence on the redox properties, including pH, the ratio of copper ions and peptide, and scan rate in electrochemical techniques. Moreover, new detection principles and technologies should be developed and the commercialization of sensing devices needs to be promoted. It is believed that more and more peptide-based biosensors will be discovered with the continuous exploration and exploration of metal complexes.

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
