# Peer review of "Progress in the Development of Biosensors Based on Peptide–Copper Coordination Interaction"

_biosensors, 2022, doi:10.3390/bios12100809_

Round 1
Reviewer 1 Report
The manuscript “Design and application of biosensors based on the peptide-copper coordination interaction” by Liu et al is a well-drafted review article that describes different types of peptides based biosensing for copper ions. The authors have extensively described the design and mechanism of reported copper detections by peptides using fluorescent, colorimetric and electrochemical methods. Overall, the manuscript is comprehensive and well written. However, I have few suggestions that might improve the outline, readability, and quality of the paper.
1. A pictorial representation summarizing the methods, model for peptide-Cu interactions, their advantages and limitations will be extremely helpful to the readers. Hence, a relevant figure (1?) can be added at the beginning of the text, which would set the tone for the readers on what might be coming in the following sections.
2. “Application” might not be the best choice of word for the title. I read through the whole manuscript hoping there are applications for these sensors in biological or material sensing, but it seemed the focus is more on the design and detection techniques mentioned above for copper and other complementary metal ion sensing.
3. For some sections, the paragraphs are very text heavy and lengthy. For e.g. for the fluorescent biosensors section (P2 and 3), the paragraphs are very long can be broken down for readers’ ease.
4. Breaking down each detection methods or “applications” into further subsection would be even better. For e.g. in fluorescent biosensors section, signal on, signal off, signal on-off-on could have been three subsections based on detection types.
5. In page 2, line 80-82, the authors mentioned about fluorescence quenching due to energy transfer to coordinated Cu2+ as a possible mechanism. Although charge transfer is predominant for such quenching, there are reports for energy transfer-based quenching. Authors should support this with proper citations.
6. Some figures are in very poor quality. For e.g. figure 6 is hard to read.
7. In figure 2, the order of A and B should be reversed for making it in line with text (caspase-3 and then B-secretase).
8. Some limit of detections are presented in molar concentration (nM or pM) where as others are presented in pg/ml (for e.g. page 8, line 300), making them harder to compare.
9. Citations required for page 8, line 304-305.
10. While colorimetric detection is one of the key methods in this manuscript, I found it is hard to quantify as compared to fluorescent or electrochemical methods. As far as I am familiar, colorimetric detections are quantified using uv-vis spectroscopy (there may be other methods!). However, there authors have not presented any UV-vis or other quantitative data for this section, like they presented for the electrochemical biosensors sections. Similar quantitative fluorescence data can be added for the fluorescent biosensors section as well.
11. For a review paper, the authors have presented too many smaller figures. Some figures in each section can be combined to make a larger figure for ease of readers.
Author Response
We thank the reviewer for his/her positive comments: The manuscript “Design and application of biosensors based on the peptide-copper coordination interaction” by Liu et al is a well-drafted review article that describes different types of peptides based biosensing for copper ions. The authors have extensively described the design and mechanism of reported copper detections by peptides using fluorescent, colorimetric and electrochemical methods. Overall, the manuscript is comprehensive and well written. However, I have few suggestions that might improve the outline, readability, and quality of the paper.
Comment 1: A pictorial representation summarizing the methods, model for peptide-Cu interactions, their advantages and limitations will be extremely helpful to the readers. Hence, a relevant figure (1?) can be added at the beginning of the text, which would set the tone for the readers on what might be coming in the following sections.
Response: It is a good suggestion. We have added scheme 1 to summarize the development of biosensors based on the peptide-copper coordination interaction.
Comment 2: “Application” might not be the best choice of word for the title. I read through the whole manuscript hoping there are applications for these sensors in biological or material sensing, but it seemed the focus is more on the design and detection techniques mentioned above for copper and other complementary metal ion sensing.
Response: We thank the reviewer for his/her comment. We have changed the title into “Progress in the development of biosensors based on peptide-copper coordination interaction.”
Comment 3: For some sections, the paragraphs are very text heavy and lengthy. For e.g. for the fluorescent biosensors section (P2 and 3), the paragraphs are very long can be broken down for readers’ ease.
Response: It is a good suggestion. We have written the long paragraphs and added sub-titles in each section.
Comment 4: Breaking down each detection methods or “applications” into further subsection would be even better. For e.g. in fluorescent biosensors section, signal on, signal off, signal on-off-on could have been three subsections based on detection types.
Response: We have written the paragraphs and added sub-titles in each section.
Comment 5: In page 2, line 80-82, the authors mentioned about fluorescence quenching due to energy transfer to coordinated Cu2+ as a possible mechanism. Although charge transfer is predominant for such quenching, there are reports for energy transfer-based quenching. Authors should support this with proper citations.
Response: We have checked and revised the presentation and cited the references.
Comment 6: Some figures are in very poor quality. For e.g. figure 6 is hard to read.
Response: We have improved the quality of the re-used figures and added the presentation in the main text to discuss the detection mechanism.
Comment 7: In figure 2, the order of A and B should be reversed for making it in line with text (caspase-3 and then B-secretase).
Response: Yes, we have revised the order of A and B in the figure.
Comment 8: Some limit of detections are presented in molar concentration (nM or pM) where as others are presented in pg/ml (for e.g. page 8, line 300), making them harder to compare.
Response: The units for the detection limits are used according to the references. “pg/ml” is used since some proteins have no precise molecular weight.
Comment 9: Citations required for page 8, line 304-305.
Response: We have cited the references to support the presentation.
Comment 10: While colorimetric detection is one of the key methods in this manuscript, I found it is hard to quantify as compared to fluorescent or electrochemical methods. As far as I am familiar, colorimetric detections are quantified using uv-vis spectroscopy (there may be other methods!). However, there authors have not presented any UV-vis or other quantitative data for this section, like they presented for the electrochemical biosensors sections. Similar quantitative fluorescence data can be added for the fluorescent biosensors section as well.
Response: In this review, we mainly discussed the design principles and analytical performances of different types of biosensors. In the electrochemical sections, a figure referring to data (Figure 7) was presented since the peptide-Cu2+ complexes exhibit a unique voltammetric behavior and electrocatalytic feature, which is highlighted in this review.
Comment 11: For a review paper, the authors have presented too many smaller figures. Some figures in each section can be combined to make a larger figure for ease of readers.
Response: In each part, the representative figures are chosen and used to depict the design principles. Other figures with similar detection principles are not included in the main text. In addition, the figures are not integrated into a larger figure in consideration of the clarity.
Reviewer 2 Report
The manuscript submitted by N. Xia and L. Liu is presenting the applications of sensors based on peptide-copper coordination interaction.
The manuscript could be accepted for publication after the authors respond to the following questions/suggestions:
General comments:
1. The subcategories of the sensors should be better emphasized (maybe by adding sub-chapters) and a comparison of the sensors from different sub-categories would be useful for the reader.
2. Some tables with the analytical performance of the described methods would be useful.
3. A graphical abstract would be useful, too.
Specific questions:
1. The introduction should better describe the novelty of the review in comparison to other reviews focused on the peptide.
The introduction should contain a short avant-premiere of the major points of the review.
2. In the abstract and introduction is suggested that the complexe peptide-Cu is used for the detection of other analytes but table 1 shoes that Cu is the only analyte.
3. Please, better organize the information, better cathegorize the information; maybe, add a figure to explain the main characteristics of different types of the sensors.
4. The reference in the text to figure 1 should be before figure 1.
5. Why does the title of Table 1 say "An overview on different sandwich biosensors for Aβ40 detection. "?
6. Other details could be added to table 1: linear range, real samples, interferences etc.
7. The resolution for figure 3 can be improved.
8. The resolution of the figure 6 should be improved.
9. Rct it is usually called charge transfer resistance not electron transfer resistance.
10. The conclusions should be more comprehensive.
Author Response
We thank the reviewer for his/her positive comments: The manuscript submitted by N. Xia and L. Liu is presenting the applications of sensors based on peptide-copper coordination interaction. The manuscript could be accepted for publication after the authors respond to the following questions/suggestions:
Comment 1: The subcategories of the sensors should be better emphasized (maybe by adding sub-chapters) and a comparison of the sensors from different sub-categories would be useful for the reader.
Response: It is a good suggestion. We have added sub-titles in each section.
Comment 2: Some tables with the analytical performance of the described methods would be useful.
Response: Three tables have been added to demonstrate the analytical performances.
Comment 3: A graphical abstract would be useful, too.
Response: We have added a scheme as the graphical abstract to summarize the development of biosensors based on the peptide-copper coordination interaction.
Comment 4: The introduction should better describe the novelty of the review in comparison to other reviews focused on the peptide. The introduction should contain a short avant-premiere of the major points of the review.
Response: It is an excellent comment. We have added the following sentences in Introduction: “Previously, several review papers have overviewed the design of peptide-derived biosensors by using peptides as the recognition elements or enzyme substrates [13-17]. However, there is little overview of the development of biosensors based on the formation of peptide-copper complexes.”
Comment 5: In the abstract and introduction is suggested that the complexed peptide-Cu is used for the detection of other analytes but table 1 shoes that Cu is the only analyte.
Response: The formed peptide-Cu2+ complexes can be further used to detect other substances that can bind to Cu2+ with high binding constant. Thus, peptides can be employed as the dual-functional probes to develop “on–off–on” biosensors for multiple analytes, which have been shown in Table 1. We have added sub-titles in each section.
Comment 6: Please, better organize the information, better cathegorize the information; maybe, add a figure to explain the main characteristics of different types of the sensors.
Response: Yes, we have added scheme 1 to summarize the development of biosensors based on the peptide-copper coordination interaction.
Comment 7: The reference in the text to figure 1 should be before figure 1.
Response: We have changed the position of Figure 1 in the main text.
Comment 8: Why does the title of Table 1 say "An overview on different sandwich biosensors for Aβ40 detection. "?
Response: We have checked the whole manuscript carefully and revised the mistakes.
Comment 9: Other details could be added to table 1: linear range, real samples, interferences etc.
Response: It is a good suggestion. We have discussed the interferences and the applications of the methods for real sample assays in the main text. However, we found that most of the references did not reported the linear ranges of the fluorescent probes.
Comment 10: The resolution for figure 3 can be improved.
Response: We have improved the resolution of the re-used figures.
Comment 11: The resolution of the figure 6 should be improved.
Response: We have improved the resolution of the re-used figures.
Comment 12: Rct it is usually called charge transfer resistance not electron transfer resistance.
Response: We have revised it.
Comment 13: The conclusions should be more comprehensive.
Response: We have revised the conclusion carefully.